# RUBATO: A MULTI-VERSION BENCHMARK FOR ROBUST MUSIC ANALYSIS AND TRANSCRIPTION

## ABSTRACT

Robustness is a fundamental challenge for deep learning, as models frequently inherit dataset biases and fail to generalize across real-world variability. Models for music audio analysis and transcription—machine-learning tasks of particular difficulty and data scarcity—often lack robustness to changes in instrumentation, interpretation, or recording conditions. In contrast to text and vision, robustness in music remains underexplored. To address this gap, we introduce RUBATO, a manually curated, fully open music dataset and benchmark. Our central idea is to exploit the unique opportunities of Western classical music where we find famous works free of copyright and with an abundance of available recordings, which follow the same score but differ in interpretation and recording conditions, supplemented by arrangements and adaptations for other instrumentations. For RUBATO, we collected and recorded 14 canonical works in up to 54 versions, totaling 560 audio tracks and 42 hours of audio, including original recordings, arrangements and adaptations, controlled piano renderings, and synthesized versions. We further curated symbolic scores and expert annotations for various tasks. Ensuring structural coherence for the majority of versions, we transfer annotations between versions using state-of-the-art alignment techniques, which we evaluate for the heterogeneous version pairs in RUBATO. The resulting high-quality annotations allow for benchmarking music understanding models, which we demonstrate for two selected tasks—automatic music transcription and local key estimation. Going beyond standard metrics, the multi-version design of RUBATO enables systematic evaluation not only of models' efficacy but also of their consistency across versions of the same work. We formalize this notion as cross-version consistency, which allows to assess model robustness along various dimensions of music data. Testing current machine-learning systems for different variants of such consistency measures, we find that most of these systems struggle to generalize under real-world variability, highlighting the need for more robust models and for benchmarks as RUBATO capable of measuring such robustness.

## 1    INTRODUCTION

Analog to the *visual* domain (Hendrycks & Dietterich, 2019), the human *auditory* system is robust in ways that current music analysis systems are not. Unlike most deep learning (DL) systems, humans can interpret music across a wide range of acoustic variations—including changes in instrumentation, interpretation, or recording conditions—without losing the ability to understand music.

Recent advances in many fields have been driven by DL and the availability of large datasets. Particularly for DL applied to music tasks, robustness—i. e., generalization across datasets and domains—remains a substantial challenge. For example, piano transcription systems that perform well on the MAESTRO dataset often show significant efficacy drops on datasets recorded under different conditions, such as MAPS (Edwards et al., 2025). Moreover, state-of-the-art models for multi-instrument transcription suffer from severe degradation in efficacy when tested on unseen datasets (Chang et al., 2024; Gardner et al., 2022).

This highlights a broader issue: current evaluation strategies rely on small or homogeneous datasets, which introduce biases in terms of instrumentation or recording conditions, among others. Without thoroughly evaluating robustness, it is hard to understand the advances in the field and to identify

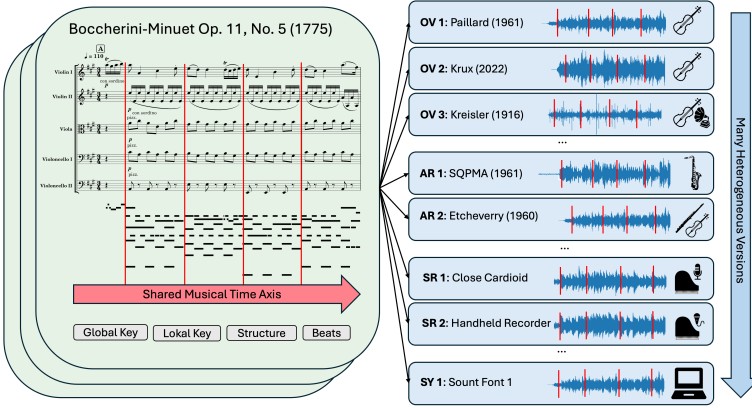

Figure 1: RUBATO dataset. For 14 famous works (*green*, see Table 1), we collected many hetero-geneous versions (*blue*, see Table 3), including original versions (OV) in differing recording quality, arrangements (AR), systematic piano renderings (SR) in different recording setups, and synthesized versions (SY)—all structurally coherent and carefully aligned to a shared musical time axis (*red*). We also provide adaptations (AD, not in figure) that diverge structurally but rely on the same works.

effective methods. This issue is particularly critical for reliable application in musicology, where biased or inconsistent model predictions can lead to distorted conclusions. While robustness is a well-studied topic for DL in vision (Hendrycks et al., 2021) and text (Niven & Kao, 2019; Elazar et al., 2021), it remains underexplored for DL algorithms applied to music. This is partly due to the limitations of existing datasets, which often focus on a single instrument as the piano in MAESTRO (Hawthorne et al., 2019) and MAPS (Emiya et al., 2010) or the guitar in GuitarSet (Xi et al., 2018) and GAPS (Riley et al., 2024)), on a single composer or work cycle such as SWD (Weiß et al., 2021), BPSD (Zeitler et al., 2024), or WRD (Weiß et al., 2023). Others only partially support open audio data (SWD, BPSD & WRD) or are not public as a whole as RWC (Goto et al., 2003), have only synthetic audio as Slakh (Manilow et al., 2019), or lack high-quality annotations and instrument balance as MusicNet (Thickstun et al. (2017), see (Gardner et al., 2022; Weiß & Peeters, 2022)).

To address this, we introduce RUBATO[1], a manually curated dataset[2] and benchmark[3] with an emphasis on heterogeneity in quality, interpretation, and instrumentation (see Figure 1). Beyond being a new resource for the data-scarce music audio domain, RUBATO unites the advantages of several existing datasets: It goes beyond the piano solo scenario and (as MusicNet but other than SWD, BPSD, WRD) features fully open audio data. As SWD, BPSD & WRD, it is a score-based multi-version data with high-quality musical annotations but, in contrast to those, is balanced across composers and work characteristics and features an extensive amount (up to 54) heterogeneous versions including systematic renderings to test specific aspects of robustness. As BPSD and SWD, the majority of versions in RUBATO exhibit a coherent structure. Going beyond all those, RUBATO combines original recordings, arrangements, systematic renderings, and synthesized audio for each work, allowing for the systematic evaluation of robustness.

As our main contributions, we (1) selected 14 famous works across 150 years of music history, featuring 12 composers and being roughly balanced in tempo, mode (major/minor), and instrumen-tation. Following the idea of (Thickstun et al., 2017), we (2) collected public domain and creative commons material (scores and recordings). Moreover, we recorded (3) several own performances in a professional studio, (4) systematic versions (MIDI reproduction piano renderings) in a controlled acoustic environment using different setups and device qualities, including video capture of the re-production piano, and (5) synthetic versions with two professional sample libraries. We (6) ensured structural coherence across most (90%) versions and (7) created high-quality musical annotations of measures, beats, instrumentation, structure, global and local keys. We (8) tested different strategies of score–audio and audio–audio alignment and (9) performed alignment with best settings to transfer annotations between score and different audio versions, obtaining a shared musical time axis.

---

[1]**R**ob**U**stness **B**enchmark for music **A**nalysis and **T**ranscripti**O**n. *Tempo rubato* (Italian for "stolen time") is an expressive tool that refers to the idea of "borrowing time" from one section of music and giving it to another.

[2]https://zenodo.org/records/17064152

[3]https://anonymous.4open.science/r/rubato-benchmark-E635

Table 1: Overview of the 14 works included in the RUBATO dataset sorted by composition year.

| ComposerID | Work ID | Title | Year | Orig. Instrumentation | # Versions |
|---|---|---|---|---|---|
| Bach | BWV1007-01 | Cello Suite Nr. 1 in G-Dur, 1. Prélude | 1724 | Cello | 30 |
| Vivaldi | RV269-01 | Le quattro stagioni: La Primavera, 1. Allegro | 1725 | Strings | 40 |
| Handel | HWV040-1-01 | Serse, Ombra mai fù | 1738 | Voice, Strings | 54 |
| Handel | HWV056-2-44 | Messiah, Hallelujah | 1767 | Choir, Orchestra | 36 |
| Boccherini | G275-03 | Quintetto d'archi, 3. Minuetto | 1775 | String Quintet | 40 |
| Beethoven | Op047-01 | Violinsonate Nr. 9 in A-Dur (Kreutzer), 1. Adagio | 1805 | Violin, Piano | 36 |
| Mozart | KV618 | Ave Verum Corpus | 1807 | Choir, Strings, Organ | 40 |
| Schubert | D733-01 | Trois Marches Militaires, 1. Allegro vivace | 1826 | Piano | 43 |
| Beethoven | Op072-0 | Fidelio, Ouvertüre | 1826 | Orchestra | 43 |
| Schumann | Op039-05 | Liederkreis, 5. Mondnacht | 1842 | Voice, Piano | 47 |
| Verdi | Nabucco-12 | Nabucco, Va pensiero sull'ali dorate | 1842 | Choir, Orchestra | 38 |
| Berlioz | H048-04 | Symphonie fantastique, 4. Marche au supplice | 1845 | Orchestra | 39 |
| Mussorgsky | Pict-10 | Pictures at an Exhibition, 10. The Great Gate of Kiev | 1886 | Piano | 52 |
| Brahms | Op115-01 | Klarinettenquintett h-Moll, 1. Allegro | 1892 | Clarinet, Strings | 22 |
| | | | | **Tracks total:** | 560 |

Beyond the dataset, we (10) propose the RUBATO benchmark, a strategy to utilize our resource as unseen data for systematically evaluating music analysis and transcription models. Going beyond standard evaluation metrics, we propose to measure consistency of model predictions across versions as an indicator for their robustness (Venohr et al., 2025; Ding et al., 2025). We (11) extend these cross-version consistency measures with two variants that measure consistency of evaluation metrics instead of model predictions. Finally, we (12) utilize these metrics for selected pairings of versions to measure robustness against specific, particularly difficult distribution shifts (e. g., from piano to other instruments or from synthesized to real audio recordings). With this strategy, we (13) compare several existing models for Automatic Music Transcription (AMT) and Local Key Estimation (LKE) and conclude on their robustness under real-world variability of music data.

## 2 RELATED WORK

To situate our research within existing literature, we summarize prior work on robustness and approaches leveraging multi-version data. We already discussed selected music datasets in Section 1.

**Robustness**: DL models are known to exploit shortcuts and inherit biases from their training data (Geirhos et al., 2020). For other domains, datasets have been proposed to benchmark model robustness to certain distribution shifts such as ImageNet-C/P (Hendrycks & Dietterich, 2019), Imagenet-R (Hendrycks et al., 2021), WILDS (Koh et al., 2021), or ARES (Liu et al., 2025). Distribution shifts can be categorized as natural or synthetic. They can be tested under worst-case scenarios, as in adversarial robustness (Goodfellow et al., 2015), or under average-case scenarios (corruption robustness, see Hendrycks & Dietterich (2019)). Since DL in music does typically not face adversarial threats, our focus is on *robustness to naturally occurring non-adversarial* distribution shifts.

**Leveraging multi-version data**: To study robustness in music analysis and transcription, Weiß et al. (2020) and Weiß & Peeters (2022) leveraged multi-version datasets by analyzing the impact of different splitting strategies. For domain adaption, Liu & Weiß (2024) used multi-version datasets within a teacher–student learning paradigm by using CVC to filter training labels. Venohr et al. (2025) and Ding et al. (2025) formalized local prediction consistency (see Section 4.1) and systematically explore it for the tasks of Multi-Pitch Estimation (MPE) and LKE. Both found this consistency not only to be a proxy for model efficacy but also to provide insight into model robustness.

## 3 RUBATO DATASET

Motivated by the limitations of existing datasets, we created RUBATO to be a heterogeneous, fully open, music dataset with high-quality annotations. We now describe its structure and content regarding works and versions (Section 3.1), alignment (Section 3.2), and annotations (Section 3.3).

### 3.1 MUSICAL WORKS AND VERSIONS

As our guiding principle, we aimed for a fully open-source dataset with performances exactly following a given musical score-based, while obtaining as many versions as possible per work. As

Table 2: Left: Track-wise instrumentation statistics of all `OV`, `AR`, and `AD` versions. Right: Note event count by instrument group in the `OV` versions.

| Instrumentation | # Tracks | hh:mm |
|---|---|---|
| Orchestra | 114 | 9:35 |
| Violin, Piano | 33 | 5:47 |
| Choir, Orchestra | 43 | 3:11 |
| Clarinet, Strings | 16 | 2:32 |
| Voice, Piano | 41 | 2:28 |
| Solo Piano | 30 | 2:02 |
| Strings | 39 | 1:57 |
| Voice, Strings | 20 | 1:04 |
| Strings, Harpsichord | 14 | 0:50 |
| Choir, Organ | 12 | 0:45 |
| Choir, Strings, Organ | 11 | 0:41 |
| Solo Cello | 15 | 0:37 |
| Solo Organ | 8 | 0:28 |
| Other | 74 | 3:48 |

| Instrument Group | # Note Events |
|---|---|
| Strings | 1 034 682 |
| Woodwind | 436 376 |
| Piano | 341 647 |
| Brass | 254 066 |
| Vocal | 60 144 |
| Harp | 3 318 |
| Percussive | 2 082 |
| Organ | 1 908 |
| Xylophone | 114 |

a consequence, we considered 12 famous composers from the common-practice period (18th & 19th century, free of copyright) and selected 14 popular works (see Table 1) with a high number of available recordings. At the same time, we strived for balance in composition years (1724–1892), instrumentation (vocal / instrumental, choir / solo voice, orchestra / chamber music, with / without piano, see Table 2), keys and modes (major / minor), as well as tempi. All remaining biases are results of our design decision for maximizing version depth.[4]

For each of the 14 works (Table 1), we collected musical scores in different formats and, as our main focus, between 22–54 audio versions per work, which we categorize into six version types (see Table 3) differing in performance style, improvisation, and degree of fidelity to the original score. Following (Weiß et al., 2021), we name all files consistently: `ComposerID_WorkID_VersionType-VersionID.ext`, for example `Boccherini_G275-03_OV-Krux2022.wav`, with the version field being left out for score-related data. We now describe our version types in detail and refer to Table 3 for an overview.

**Score versions**: We collected open-source scores, applied OMR if necessary, corrected the machine-readable scores in MuseScore (an open source editor), and adapted them to scholarly-critical editions. Along with MuseScore files, we exported image, PDF, MusicXML, and MIDI versions, the latter serving for generating note-level annotations and for rendering systematic piano recordings.

**Real-world audio versions**: To stick with open audio material, we manually collected professional recordings, which are in the Public Domain[5] or released under Creative Commons licenses. In addition, we record additional versions of four works including `Brahms_Op115-01`, all featuring conservatory students and graduates as performers. Among the different audio versions, we have

- *Original Versions* (`OV`), which exactly follow the original score in terms of instrumentation and structure, thus containing the exactly same note events up to temporal variations. One challenge are structural differences between versions such as repetitions played in some and ommitted in other versions. Following (Zeitler et al., 2024), we manually edit recordings deviating from our reference original (`OV-R`) to ensure structural coherence. Additionally, we annotate all transpositions and octave shifts of vocal soloists (occurring when comparing female and male voices).
- *Arrangements* (`AR`), which preserve the overall structure and harmony but may differ in instrumentation and exact pitch content, especially octave/register of notes (e. g., a guitar version of the *Cello Suite*). For `Mussorgsky_Pict-10`, along with the 13 original piano versions, we have 21 orchestral arrangements following M. Ravel's orchestration.
- *Adaptations* (`AD`), which may significantly deviate from the original either in structure (`AD-S`) or in both structure and instrumentation (`AD`). The original work remains recognizable. This category includes unique renditions such as a barrel-organ version of `Verdi_Nabucco-12`. These versions can be used for global tasks, such as version retrieval, or global key estimation.

---

[4]Please note that RUBATO serves to evaluate and improve *technical* methods for music analysis, for which cultural and gender biases are not expected to be relevant. The resulting methods can then be used for *musicological* studies addressing such biases by deliberately considering female, non-white composers, and non-Western musical cultures. For these reasons, we prioritized acoustic/musical diversity over diversity of creators.

[5]Under EU law, performances recorded before 1963 are in the Public Domain, which we used as basis for inclusion. Please note that these recordings might not be in the Public Domain in countries outside the EU.

Table 3: Version types in the RUBATO dataset. Possible tasks include transcription+instrument (1), beat tracking (2), LKE (3), pitch class activity (4), chord (5, will be added), structure analysis (6), instrument detection (7), global key estimation(8) and version retrieval (9).

| Version Type | Struct. | Pitch | Instr. | Alignment | Tasks | # Vers. | hh:mm |
|---|---|---|---|---|---|---|---|
| Original Ref. Version (OV-R) | ✓ | ✓ | ✓ | manual | 1-9 | 14 | |
| Original Version (OV) | ✓ | ✓ | ✓ | transferred | 1-9 | 269 | 24:40 |
| Ref. Arrangement (AR-R) | ✓ | (✓) | ✗ | manual | 3-9 | 10 | |
| Arrangement (AR) | ✓ | (✓) | ✗ | transferred | 3-9 | 122 | 7:48 |
| Adaptation (AD-S) | ✗ | (✓) | ✓ | - | 7-9 | 36 | |
| Adaptations (AD) | ✗ | (✓) | ✗ | - | 7-9 | 19 | 3:20 |
| Systematic Rendition (SR) | ✓ | ✓ | ✗ | exact | 1-9 | 60 | 4:29 |
| Synthesized (SY) | ✓ | ✓ | ✓ | exact | 1-9 | 30 | 2:16 |
| | | | | | | 560 | 42:36 |

**Controlled audio versions**: In addition, we created controlled versions to serve as a standardized reference across all works, to ensure comparability among works and to enable investigation of particular domain shifts (such as between piano–orchestra or synthetic–real audio). For the *Systematic Renderings* (SR), we recorded each work on a MIDI-controllable Yamaha C3X ENSPIRE PRO grand piano in an acoustically optimized studio (see Appendix A.2) with multiple microphone setups for comparability: close-miking and room-miking (both using Schoeps MK5 microphones), as well as a consumer-grade handheld stereo recorder (Zoom H4n) to account for low-fidelity conditions. Starting from the original score (e. g., for orchestra), we extract a MIDI file, map all note events to one channel, and clean up notes events (merging duplicates, splitting overlapping events). We captured recordings via a Yamaha DM3 digital mixing desk using Nuendo software, and exported both with and without reverb. Additionally, we provide two *Synthesized Versions* (SY), rendered from symbolic scores using high-quality sampling libraries: East West Symphonic Orchestra (EWSO) and Steinberg HALion Symphonic Orchestra (HSO). We provide all recordings mono WAV files at 22.05 kHz sampling rate to be used for research purposes.

**Other Versions**: In addition to the audio versions, we also include synchronized video recordings of the automatically moving piano keys to be used, e. g., for the task of visual piano transcription (Koepke et al., 2020).

### 3.2 CROSS-VERSION ALIGNMENT STRATEGY

Different versions of a work generally follow the same score regarding pitch information (note sequence), but have large freedom in shaping global and local tempo including fluctuations such as agogics, ritardando, or rubato. To align the physical timelines across all versions and with the musical timeline of the score, we employ a multi-stage synchronization approach inspired by prior work (Weiß et al., 2021; 2023; Zeitler et al., 2024).

**Manual annotations:** For each work, we selected two reference versions, one original (OV-R) and one arrangement (AR-R) and manually annotated downbeat (measure) positions, for OV-R also beat positions using Sonic Visualizer (Cannam et al., 2010).

**Annotation transfer:** We then used audio–audio synchronization to transfer the manual annotations to all other OV/AR versions. The transferred beat annotations serve as anchor points for a subsequent fine-grained score–audio alignment. This step aligns the score's musical timeline with each version's physical timeline, enabling precise transfer of note events and other annotations.

**Alignment quality**: For synchronization, we used the SyncToolbox (Müller et al., 2021) implementation of memory-restricted multi-scale dynamic time warping (MrMsDTW) (Prätzlich et al., 2016). While the signal processing (SP) features of the SyncToolbox are known to obtain high quality for audio–audio synchronization (especially with piano), our heterogeneous pairings scenarios (score–audio, synthetic–real, piano–orchestra, high–low quality) may pose challenges. To assess the quality of our alignment in these scenarios, we conducted an in-depth study using our manual measure annotations (for OV-R/AR-R) and systematic versions (SR, SY), where we obtain measure annotations from the source MIDI file. We compared various SP features as used by Müller et al. (2021) with DL-based pitch and chroma features derived from a MPE model (Weiß & Peeters, 2022). From the results (shown in Appendix A.3), we find that DL-based chroma features, especially in combination

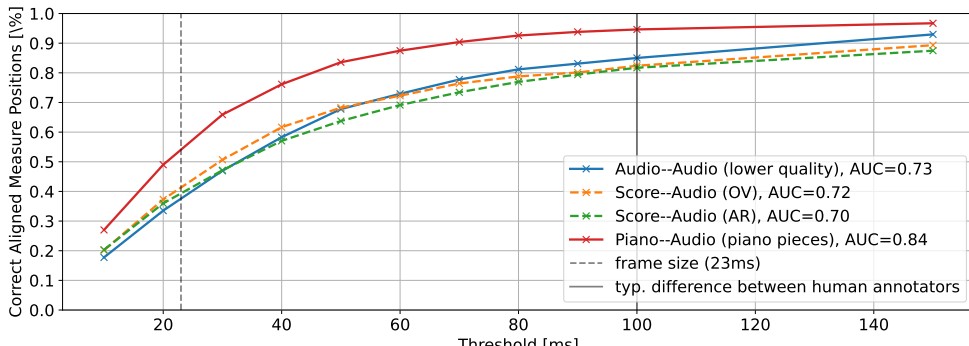

Figure 2: Alignment accuracy curves averaged over manually annotated versions. The curves show the percentage of correctly aligned measure positions within a given tolerance.

with SP-based onset features, achieve higher accuracy than other features such as SP-based chroma or pitch vectors. Based on these findings, we adopted this hybrid feature approach (DL-based pitch class features combined with SP-based chroma onset estimates) with a frame rate of 43.07 Hz.

Figure 2 shows accuracy curves with these features for various heterogeneous pairings. The alignment of `SR` versions to other piano versions, a homogeneous scenario, achieves the highest accuracy (AUC = 0.84). Our primary scenarios for this dataset (score–audio and audio–audio with different recording quality), still achieve high accuracy despite considerable heterogeneity of the data. To contextualize the accuracy of our alignment, we need to relate the values to human cross-annotator consistency, which has been shown to be in the order of 100 ms and higher for complex classical works such as romantic operas Weiß et al. (2016). As seen in Figure 2, at this 100 ms tolerance, we achieve over 80% in heterogeneous and up to up to 95% in homogeneous cases. For the quality of the resulting annotations, these curves rather indicate lower bounds, since we use additional anchor points (from manual measure and beat positions) to support the annotation transfer. For the reference versions (manually annotated), the quality is naturally even higher, and the MIDI-based `SY` and `SR` versions have perfect annotations derived from the score.

### 3.3 MUSICAL ANNOTATIONS

RUBATO provides a variety of expert-created musical annotations, which may serve to evaluate and improve a range of music analysis and transcription tasks (compare Table 3). As described above, we obtain beat and downbeat (measure) annotations for all version. Based on the score, we manually create structure and local key annotations. Relying on our measure/beat positions and alignments (see Section 3.2), we transfer these annotations to all versions (excluding `AD`). Similarly, we derive note-level annotations (including instrument labels and lyrics) for all `OV` versions. We further provide track-wise annotations such as the global key and the overall structure. We organize annotations into folders and name annotation files in accordance with the reference audio files.

## 4 RUBATO BENCHMARK

Like any dataset, RUBATO can be splitted for train–test experiments. For this case, we propose a best-practice split strategy in Appendix A.5.2. However, we primarily conceive RUBATO as an unseen benchmark, designed to expose robustness gaps in music analysis systems and, therefore, recommend usage as a hold-out test set. What makes RUBATO distinctive is its structure and its heterogeneity: for each work, multiple aligned audio versions are available, differing in performers, instrumentation, and recording conditions. This diversity makes RUBATO not only a challenging benchmark, but also uniquely suited for studying robustness. In particular, it allows us to ask:

> *How consistent are model predictions across different versions of the same work,*
> *i. e., when musicians, instruments, or recording conditions change?*

To address this question, we introduce several robustness measures supplementing standard metrics and use them to benchmark pre-trained models for AMT and LKE.

## 4.1 Cross-Version Consistency

Existing evaluation metrics measure how well models analyze or transcribe individual recordings, but fail to capture whether model predictions are robust, i.e., remain stable under domain shifts. In practice, transcription systems often do well on one version of a work but fail on another, even though the underlying musical content is identical (see, e.g., Figure 3).

For musicological applications, such inconsistencies are problematic. Comparative studies like corpus analysis require model robustness to performer, instrument, and recording changes. To explicitly measure this robustness, we introduce Cross-Version Consistency (CVC, see (Ding et al., 2025; Venohr et al., 2025))—an additional figure of merit that quantifies how consistent model predictions remain across different versions of the same work. We define three notions of CVC: **Global Evaluation Consistency (GEC)** measures how much track-wise model efficacy (given an evaluation metric) differs between version of a work. Visually, this corresponds to the average pairwise vertical proximity per work in Figure 3. **Local Evaluation Consistency (LEC)** goes beyond GEC as it additionally considers *when* errors occur by locally comparing efficacy at corresponding frames. **Local Prediction Consistency (LPC)** goes beyond LEC as it captures *which* errors occur by directly comparing model predictions at corresponding frames. As LPC is annotation-free, it is particularly relevant for tasks where annotations can be ambiguous (e.g. LKE) or inaccurate (e.g. AMT).

**Version pairs**: Let each audio version $X_{w,v}$ be uniquely defined by a work $w$ and a version $v$. We define the set of all cross-version pairs as $\mathcal{P} = \{(X_{w_1,v_1}, X_{w_2,v_2}) \mid w_1 = w_2, \ v_1 \neq v_2\}$. Let $(X_1, X_2) \in \mathcal{P}$ be a cross-version pair and $Y_1, \hat{Y}_1 \in \mathbb{R}^{N \times B}$ and $Y_2, \hat{Y}_2 \in \mathbb{R}^{M \times B}$ their respective musical annotations and model predictions, with $B$ being the number of output dimensions (e.g. 88 pitch bins for MPE or 24 key classes for LKE) and $N$ and $M$ being the number of time frames at a given frame rate. We compute CVC for all pairs in $\mathcal{P}$ or for certain subsets and then average.

**GEC**: Given a global evaluation measure $g : \mathbb{R}^{N \times B} \times \mathbb{R}^{N \times B} \to [0, 1]$ we define GEC $\in [0, 1]$ as:

$$\mathrm{GEC}(X_1, X_2) = 1 - \left| e(\hat{Y}_1, Y_1) - e(\hat{Y}_2, Y_2) \right|. \tag{1}$$

**Local comparison**: As all version pairs share a common musical time axis, we can define musically meaningful local comparison. Based on beat annotations (Section 3.2), we derive a warping path $P = \big( p(1), \dots, p(L) \big), \quad p(l) = (n_l, m_l) \in [1 : N] \times [1 : M]$ with $L \geq \max(N, M)$ that establishes correspondences between time frames of two sequences of lengths $N$ and $M$.

**LEC**: Given an local evaluation measure $e : \mathbb{R}^B \times \mathbb{R}^B \to [0, 1]$, we define LEC $\in [0, 1]$ as:

$$\mathrm{LEC}(X_1, X_2) = 1 - \frac{1}{L} \sum_{l=1}^{L} \left| e\big(\hat{Y}_1(n_l), Y_1(n_l)\big) - e\big(\hat{Y}_2(m_l), Y_2(m_l)\big) \right|. \tag{2}$$

**LPC**: Given suitable similarity measure $s : \mathbb{R}^B \times \mathbb{R}^B \to [0, 1]$, we define LPC $\in [0, 1]$ as:

$$\mathrm{LPC}(X_1, X_2) = \frac{1}{L} \sum_{l=1}^{L} s(\hat{Y}_1, \hat{Y}_2). \tag{3}$$

As some versions are performed in different keys, we transpose $\hat{Y}$ accordingly before computing the similarity. While, by definition, a model making constant but incorrect predictions (e.g., predicting the same pitch for every frame) yields a high LPC score, we note that this does not invalidate the metric since we consider CVC as an *additional figure of merit* rather than a standalone metric.

**Aspects of Consistency**: To gain a nuanced insight into different aspects of robustness, we define subsets of $\mathcal{P}$ to make certain domain shifts explicit. "Piano–Other" measures CVC on all version pairs of SR and OV and thus the shift from rendered piano recordings to real-world recordings of various instrumentations. "Synthetic–Real" aggregates CVC between all pairs of SY and OV, thus capturing the shift from synthetic renderings to real-world recordings with the same instrumentation.

## 4.2 Experiment 1: Automatic Music Transcription

We now want to conduct our benchmark for two selected task. In the first experiment, we evaluate several multi-instrument AMT models for their efficacy and CVC on RUBATO.

**Evaluation:** We assess model efficacy using both note-level and frame-level metrics. At the note level, we report $F_{\mathrm{On}}^{\Delta}$, a pitch-onset F-measure that takes note as correct if its pitch is within a quarter tone and its onset within $\pm\Delta$ ms of the reference. We also report a pitch-wise onset/offset measure

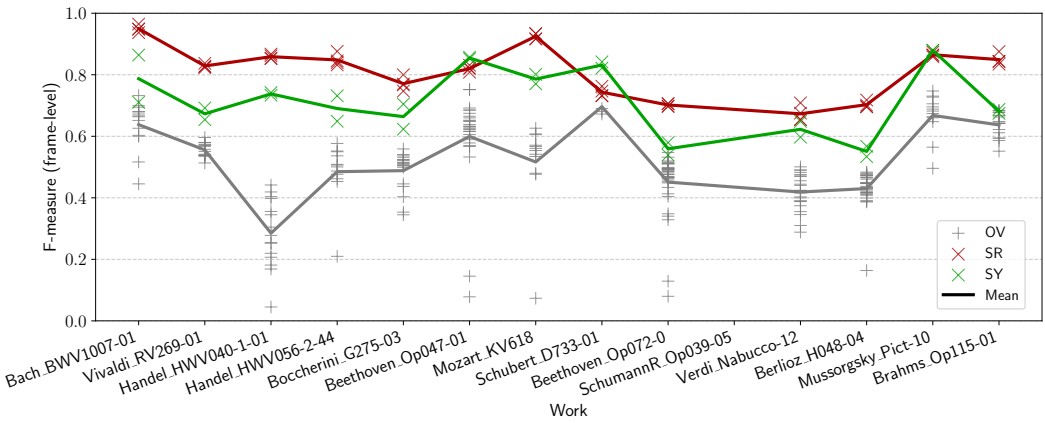

Figure 3: Frame-level AMT results per work for `MT3` (see Appendix B for other models).

Table 4: RUBATO benchmark for two AMT tasks. Results for six selected models.

| Task | Model | Standard Evaluation Metrics | | | | | | Cross-Version Consistency Measures | | | | | | | | |
| | | Note-Level | | Frame-Level | | | | All Pairs | | | Piano–Other | | | Synth–Real | | |
| | | $F_{On+Off}^{100}$ | $F_{On}^{100}$ | P | R | F | AP | GEC | LEC | LPC | GEC | LEC | LPC | GEC | LEC | LPC |
|---|---|---|---|---|---|---|---|---|---|---|---|---|---|---|---|---|
| MPE | ResNet | - | - | **76.9** | **63.4** | **68.3** | **75.0** | **93.8** | **77.5** | **65.1** | **89.6** | **73.6** | **56.7** | **90.6** | **74.6** | **61.2** |
| | BP-FRAME | - | - | 41.0 | 39.9 | 38.9 | 32.8 | 90.7 | 77.1 | 41.5 | 89.4 | 73.3 | 34.8 | 85.5 | 72.6 | 39.2 |
| Note | BP-NOTE | 20.1 | 39.5 | 73.2 | 44.5 | 51.9 | - | 84.5 | 70.5 | **44.3** | 78.4 | 65.1 | 38.1 | 75.6 | 64.3 | **41.7** |
| | ReconVAT | 17.7 | 35.7 | **73.9** | 26.9 | 37.3 | - | 89.2 | **72.0** | 35.7 | **86.0** | **66.7** | 26.3 | **81.3** | **66.7** | 31.7 |
| | MT3 | **35.1** | **51.8** | 69.5 | **50.3** | **57.0** | - | 84.8 | 66.1 | 41.5 | 71.1 | 59.1 | **40.5** | 79.1 | 62.4 | 39.3 |
| | YMT3+ | 32.2 | 49.7 | 39.3 | 28.8 | 32.0 | - | 77.2 | 65.4 | 25.2 | 49.2 | 42.6 | 19.9 | 79.8 | 65.8 | 22.5 |

$F_{On+Off}^{\Delta}$, which further requires the offset to be within $\pm\Delta$ ms or 20% of the reference note's duration. We match reference and predicted notes using `mir_eval` (Raffel et al., 2014). Since annotation accuracy is lower in multi-instrument scenarios (see Section 3.2), we set $\Delta = 100$ ms. At the frame level, we compute Precision (P), Recall (R), and F-measure by comparing binary pianoroll representations of predictions and targets in $\{0, 1\}^{N \times 72}$ at a frame rate of 43.07 Hz. For models with probabilistic outputs, we additionally report Average Precision (AP). For CVC, we binarize outputs and choose F-measure as metric for $e$, $g$, and $s$. By definition, we evaluate CVC on the frame-level.

**Models:** We evaluate several pre-trained models designed for frame-level, note-level, or hybrid transcription. For frame-level AMT results, we use a medium-sized `ResNet` (Weiß & Peeters, 2022), trained on other classical music multi-version datasets as described by Venohr et al. (2025). As a hybrid model, we test the Notes and Multipitch (NMP) model (Bittner et al., 2022), referred to as Basic Pitch (BP). We evaluate both its frame-level output $Y_n$ before decoding (`BP-FRAME`) and its note-level output after decoding (`BP-NOTE`). For pure note-level models, we consider `ReconVat` (Cheuk et al., 2021), trained in a semi-supervised fashion, as well as two Transformer-based models that directly output note events: `MT3` (Gardner et al., 2022) and `YMT3+` (Chang et al., 2024).

**Results**: Figure 3 shows frame-level efficacy of `MT3`. As expected, the controlled piano renderings (`SR`) are easiest to transcribe and generally provide an upper bound for each work. Efficacy decreases for synthetic (`SY`) and further for real-world recordings (`OV`), with some versions failing entirely. Looking at Table 4, we can see that note-level metrics are generally low: even with a high tolerance and ignoring offsets, the best $F_{On}^{100}$ is 51.8 for `MT3`. This highlights the challenge of note-level AMT in this multi-instrument setting. Interestingly even though `MT3` and `YMT3+` perform similar on the note-level, `MT3` is better on the frame-level (F = 57 vs. 32). Looking at the consistencies, we find that `YMT3+` is also the most inconsistent among the tested models. It seems to be particularly inconsistent between piano and other instrumentations. Summarizing all CVC measures, we find `BP-NOTE` as note-level model and `ResNet` for MPE to obtain highest consistencies. This suggests that medium-sized, fully convolutional models obtain slightly lower efficacies but are more robust and, therefore, universally applicable than large Transformers.

### 4.3 EXPERIMENT 2: LOCAL KEY ESTIMATION

As our second experiment, we benchmark LKE. As an essential component of harmony, LKE suffers from inherent musical ambiguity and high annotation subjectivity. Weiß et al. (2020) showed that

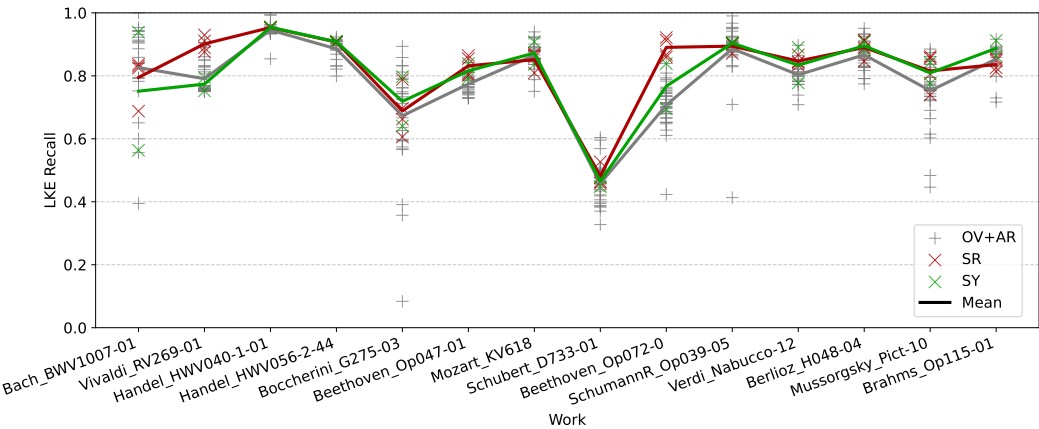

Figure 4: Key recall per work for `Octave-LSTM`. Since the `AR` versions keep the harmony content, we evaluate on the `AR` versions as well.

Table 5: Rubato benchmark for LKE. Results for two selected models.

| Task | Model | Standard Metrics | | Cross-Version Consistency Measures | | | | | | | | |
| | | | | All Pairs | | | Piano–Other | | | Synth–Real | | |
| | | Recall | MIREX | GEC | LEC | LPC | GEC | LEC | LPC | GEC | LEC | LPC |
| LKE | CNN | 66.7 | 73.5 | **93.3** | 82.1 | 70.7 | **92.4** | 80.3 | 71.4 | 92.7 | 81.8 | 72.4 |
| | Octave-LSTM | **79.7** | **85.2** | 93.2 | **88.8** | **85.4** | 91.9 | **86.6** | **83.0** | **93.7** | **89.2** | **85.1** |

inter-annotator agreement can be as low as 75% for this task. However, current DL models often achieve recall rate higher than 80%, when measured against one single annotation, suggesting that these models might have overfitted to certain annotators. Therefore, Ding et al. (2025) proposed to use CVC as an annotation-free evaluation strategy for LKE, avoiding the bias introduced by annotations. We follow this prior work and evaluate two models on RUBATO.

**Evaluation:** We evaluate models' efficacy using two different standard metric for key estimation. The first one is key recall rate, which is the accuracy ignoring frames annotated as "no key." The second metric is MIREX score, which gives partial scores to fifth errors, relative errors, and parallel errors. We use the `mir_eval` package (Raffel et al., 2014) to compute the MIREX score.

**Models:** We consider two baseline models from Ding et al. (2025). The first one is a fully convolutional network (`CNN`), the second one is a CRNN with octave-based rearrangement (`Octave-LSTM`). We train the models on other cross-version datasets as in Ding et al. (2025).

**Results**: Figure 4 shows the key recall rate for `Octave-LSTM`. Efficacy is generally high as compared to a typical inter-annotator agreement. In contrast to AMT, results on different version types are similar, and using systematic rendering or synthetic data does not make LKE easier. The observation with `CNN` are similar. This suggests that for LKE, the challenge is less the variety of versions but to learn the *musical notion* of local key. Looking at the cross-version consistencies (Table 5), we find that `Octave-LSTM` achieves higher recall and MIREX score, and is also more consistent across different versions, with higher LEC and LPC values. We note that GEC values do not correspond to efficacy—both models achieve similar GEC but `Octave-LSTM` obtains higher recall. Therefore, we argue that global consistencies alone as used by Weiß et al. (2020) are insufficient since models can make different mistakes in different versions, which may balance out globally. Thus, we consider local metrics (LEC and LPC) necessary to understand model robustness.

## 5 CONCLUSIONS

With RUBATO, we contribute the first openly available, systematic multi-version dataset explicitly designed to study robustness in music analysis and transcription. Its 560 tracks comprising heterogeneous versions of 14 works from Western classical music span a wide range of instruments, performers, and recording conditions, thus offering a unique opportunity to benchmark ML model behavior under real-world variability. Beyond the exemplary results shown in this paper, the RUBATO benchmark enables to evaluate a variety of further tasks such as beat, downbeat, or structure analysis, and we plan to enrich the RUBATO dataset with further annotations in the future.

ACKNOWLEDGMENTS

— anonymized for double-blind review —

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

APPENDIX

# A RUBATO DATASET DETAILS

In the following subsections, we provide further details on the dataset creation process and its structure.

## A.1 BACKGROUND

Despite some recordings created by the authors (see Section A.2), the majority of data in RUBATO is based on public sources, released under Creative Commons licenses or being in the Public Domain due to copyright expiry (including performer rights). The public sources includes symbolic score data, which we manually revised using MuseScore, including the correction of OMR errors, verification of note durations, pitches, and time signatures. Moreover, it comprises a high number of audio files.

A key aspect of RUBATO is that the temporal structure—specifically, the number and ordering of measures—remains consistent across all score and audio versions (except AD). To achieve this, following the strategy by Zeitler et al. (2024), we either adapt the scores by removing repetitions or edit the audio tracks (by cutting repeated sections if necessary) to align with the score structure. This ensures that the largest possible number of tracks shares the same overall structure. For some versions, this is not possible since they leave out parts that are not played elsewhere (e. g., when a first ending is missing due to the repetition being left out). In such cases, we do not edit the recordings but sort them into the structurally deviating adaptations (AD-S). Finally, we resample all tracks to 22,05 kHz and export them as mono WAV files.

Table 6 shows the number of versions for each work regarding the different version types. Note that for *The Great Gate of Kiev* from *Pictures at an Exhibition* (Finale), we captured systematic renderings and synthesized audio both for Mussorgsky's original piano version and for Ravel's orchestration, which is our most frequent arrangement for this work. For orchestral and larger chamber works, arrangements with alternating instrumentation are hard to find under our open-access constraints.

For each work, we selected a single OV-R version to serve as reference for annotation. While we tolerate interpretative freedom in terms of tempo and articulation, the performance must preserve the overall structure and be of good audio quality compared to other versions of this work. As a counterpart, we selected a reference arrangement version (AR-R) as well, intended to diverge from the OV-R in terms of instrumentation and musical interpretation while maintaining structural integrity.

Afterwards, a single musically trained annotator manually created high-quality measure (for OV-R and AR-R) and beat annotations (for OV-R only). Due to the ambiguity involved even in this task (Weiß et al., 2016), these annotations should be interpreted as a consistent reference rather than an absolute "truth." Since most real-world recordings in RUBATO date from the first half of the 20th century, audio quality varies strongly. In particular, low-quality AR-R versions sometimes made it difficult to identify clear measure onsets, especially in multi-instrument passages or when notes extended across measure boundaries. In such cases, we placed measure positions on the clearest audible note change or inferred them based on the tempo and rhythmic context. For details on the transfer of these annotations to the remaining versions, please see Section A.4.

## A.2 RECORDING

In the following, we provide more details on the recording process. For many musical styles such as folk music of various cultural traditions or Western classical music, the quality and characteristics of recordings may vary to a high degree. Artifacts of old recording devices (shellac, vinyl, tape) and the subsequent digitization process, the used recording devices (microphones, AD converters, etc.), and acoustic conditions of the performance (such as reverb and frequency response of the recording space, or the concrete instruments used) play a role. To study these characteristics, we created a number of systematic audio versions in an acoustically optimized studio with professional and consumer-grade audio equipment and an expert sound engineer.

Table 6: Number of version types per work. * `Mussorgsky_Pict-10` provides `SR` and `SY` both for the piano original and an orchestra version.

| Work ID | Title | Global Key | # OV | # AR | # AD | # SR | # SY | # Total |
|---|---|---|---|---|---|---|---|---|
| BWV1007-01 | Cello Suite Nr.1 in G-Dur BWV10007, 1. Prélude | G:maj | 15 | 7 | 2 | 4 | 2 | 30 |
| RV269-01 | Le quattro stagioni: La Primavera, 1. Allegro | E:maj | 16 | 18 | - | 4 | 2 | 40 |
| HWV040-1-01 | Serse, Ombra mai fù | F:maj | 14 | 23 | 11 | 4 | 2 | 54 |
| HWV056-2-44 | Messiah, Hallelujah | D:maj | 12 | 12 | 6 | 4 | 2 | 36 |
| G275-03 | Quintetto d'archi, 3. Minuetto | A:maj | 19 | 10 | 5 | 4 | 2 | 40 |
| Op047-01 | Violinsonate Nr. 9 in A-Dur (Kreutzer), 1. Adagio | A:min | 28 | - | 2 | 4 | 2 | 36 |
| KV618 | Ave Verum Corpus KV618 | D:maj | 12 | 13 | 9 | 4 | 2 | 40 |
| D733-01 | Trois Marches Militaires, 1. Allegro vivace | D:maj | 4 | 25 | 8 | 4 | 2 | 43 |
| Op072-0 | Fidelio, Ouvertüre | E:maj | 37 | - | - | 4 | 2 | 43 |
| Op039-05 | Liederkreis, 5. Mondnacht | E:maj | 39 | 2 | - | 4 | 2 | 47 |
| Nabucco-12 | Nabucco, Va pensiero sull'ali dorate | F#maj | 25 | 1 | 6 | 4 | 2 | 38 |
| H048-04 | Symphonie fantastique, 4. Marche au supplice | G:min | 33 | - | - | 4 | 2 | 39 |
| Pict-10 | Pictures at an Exhibition, 10. The Great Gate of Kiev | Eb:maj | 13 | 21 | 6 | 8* | 4* | 52 |
| Op115-01 | Klarinettenquintett h-Moll, 1. Allegro | B:min | 16 | - | - | 4 | 2 | 22 |
| | | | 283 | 132 | 55 | 60 | 30 | 560 |

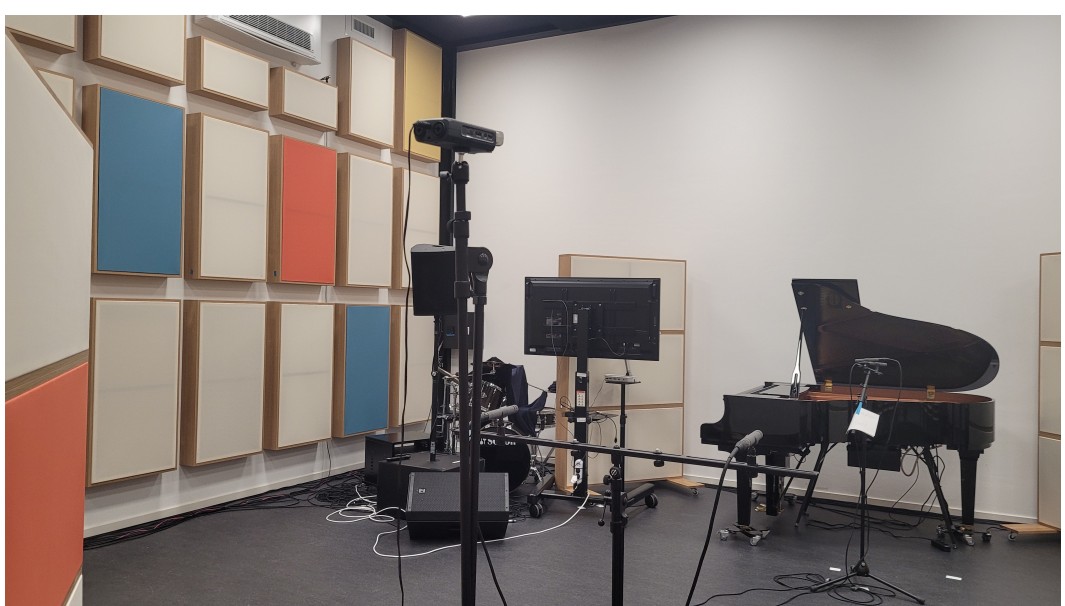

Figure 5: MIDI-controllable reproduction piano (Yamaha C3X Enspire Pro) in the acoustically optimized studio with different microphone setups. Handheld recorder, AB-microphones for room sound and ORTF microphones for close-miking.

Figure 5 shows a photograph of the studio and the recording setup. The room size is 9×9 m with 4.5 m height. Walls (except stage side as live end) and ceiling are optimized with diffusors and absorbers (material: Caruso Isobond), with additional movable diffusor/absorber walls. This results in a nearly flat frequency response above 200 Hz with a reverb time (RT60) of roughly 0.5 s in the empty room (see Figure 6). These conditions guarantee for a neutral recording scenario.

Besides the acoustic and recording conditions, musical variations originating from the performer's interpretative freedom constitute a central source of variability between versions. To disentangle these performance aspect from the acoustic characteristics, we create systematic renderings. To this end, we make use of the playback capabilities of a MIDI-controllable reproduction piano, a Yamaha C3X Enspire Pro grand piano (Figure 5). Yamaha Enspire is the successor technology of the Yamaha Disklavier, which allows to convert MIDI signals into precise mechanical actions on the piano. The process is as follows: We start from the high-quality symbolic scores curated in MuseScore (Section 3.1) of the original versions (full scores). Via MusicXML, we import these into professional notation software (Steinberg Dorico), which allows us to export MIDI files with full control over the velocities. We use these MIDI files to export two synthetic audio versions using two professional

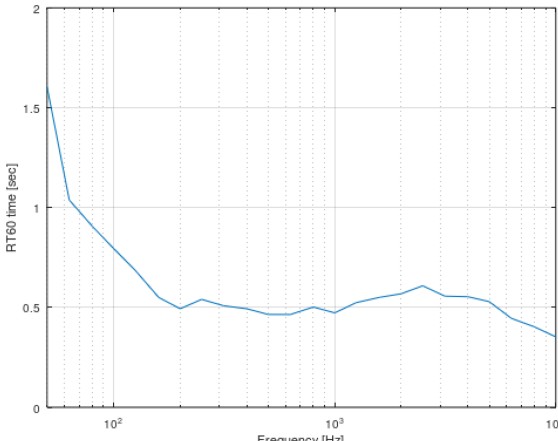

Figure 6: Frequency-dependent reverb time (RT60) of the empty recording studio, averaged over six microphone positions across the room.

Table 7: Audio versions captured from the systematic piano renderings (SR)

| VersionID | Microphones | Distance | Technique | Post-processing |
|---|---|---|---|---|
| SR-R-ReproPno-CloseRev | Schoeps MK5 cardioid | close (1 m) | ORTF | EQ, Reverb |
| SR-ReproPno-CloseDry | Schoeps MK5 cardioid | close (1 m) | ORTF | EQ |
| SR-ReproPno-RoomRev | Schoeps MK5 omnidirectional | far (5 m) | AB | EQ |
| SR-ReproPno-HandRecDry | Zoom H4N, stereo condenser | far (5 m) | XY | – |

sample libraries (East West Symphonic Orchestra, EWSO, and Steinberg HALion Symphonic Orchestra, HSO). Next, we slightly modify the output MIDI files using the PrettyMIDI Python library to scale velocities into a range that musical experts considered adequate for the notated dynamics (piano, fortissimo etc.). Moreover, this script sets all channels to piano and marginally shortens some note events to guarantee enough time ($\geq 10$ ms) in between successive activations of the same piano key. The final MIDI files are the ones provided in the RUBATO dataset; timing follows constant tempo and all time positions are consistent across the MIDI and all SR and SY versions, as well as the captured video.

We record this rendering process with multiple setups simultaneously. As our central recording device, we use a Yamaha DM3 digital mixing desk connected to a Digital Audio Workstation (DAW, Steinberg's Nuendo). We capture the piano signal with two microphone setups, both using professional Schoeps MK5 condenser microphones: one pair (close) captures the piano signal from roughly 1 m distance with cardioid characteristic in ORTF setup, the other pair captures the room signal from roughly 5 m distance with omnidirectional characteristic in AB setup. After balancing slight colorations through equalization (EQ), we export both audio files in dry conditions. For the ORTF signal, we also create an artificially reverberated version using a standard algorithmic reverb plugin of the DAW. We capture all signals at 48 kHz and downsample them to 22.05 kHz mono afterwards for consistency with other versions (stereo recordings are available upon request). To account for a consumer-grade recording, we independently record with a handheld recorder (Zoom H4n) with direct MP3 compression (bitrate 256 kBit/s), manually correcting for a global time change compared to the other recordings afterwards. Moreover, we also capture a video recording of the moving piano keys and sustain pedal using a consumer-grade webcam. These multiple versions of the exact same mechanical performance allow for systematically comparing algorithms across various conditions (dry–reverb, high–low quality, close–far) and modalities (audio–video–MIDI). Table 7 provides a systematic overview of the conditions.

Finally, using the same room and recording setup, we also record own versions of selected works with trained, semi-professional performers (involving the same piano but played by humans).[6] These include:

---

[6]For double-blind review, we mask the performer names by replacing VersionIDs with XXXX.

Table 8: Audio Pairings used for our feature comparison study.

| Work | Version 1 | Version 2 | Instrumentation 1 | Instrumentation 2 |
|------|-----------|-----------|-------------------|-------------------|
| Bach_BWV1007-1 | OV-R-Fournier1961 | AR-R-Oribe | Cello | Guitar |
| Beethoven_Op047-01 | OV-R-Szeryng | OV-Taschner1951 | Violin, Piano | Violin, Piano |
| Beethoven_Op072-0 | OV-R-Jochum1960 | OV-Karajan1960 | Orchestra | Orchestra |
| Berlioz_H048-04 | OV-R-Markevitch1953 | OV-Paray1960 | Orchestra | Orchestra |
| Boccherini_G275-03 | OV-R-Paillard1961 | AR-R-Suarez2023 | Strings | Piano |
| Brahms_Op115-01 | OV-R-McColl1988 | OV-Draper1933 | Clarinet, Strings | Clarinet, Strings |
| Handel_HWV040-1-01 | OV-R-Schlusnus1923 | AR-R-Bra2020 | Voice, Strings | Accordions |
| Handel_HWV056-2-44 | OV-R-Ormandy1959 | AR-R-Soderos1915 | Choir, Orchestra | Marching Band |
| Mozart_KV618 | OV-R-Arndt1962 | AR-R-Tung2013 | Choir, Strings | Flutes |
| Mussorgsky_Pict-10 | OV-R-Staab2016 | AR-R-Karajan1957 | Piano | Orchestra |
| Schubert_D733-01 | OV-R-BouchardMorisset | AR-R-Gruber1960 | Piano | Orchestra |
| Schumann_Op039-05 | OV-R-Jurinac1954 | AR-R-SchumannER1923 | Voice, Piano | Voice, Orchestra |
| Verdi_Nabucco-12 | OV-R-Kempen1951 | AR-R-Operaphilia2022 | Choir, Orchestra | Choir, Piano |
| Vivaldi_RV269-01 | OV-R-Gawriloff1961 | AR-R-Intartaglia2011 | Strings | Organ |

- `Schubert_D733-01_OV-XXXX.wav`: An original version of Schubert's four-hand piano military march.

- `Schumann_Op039-05_OV-XXXX.wav`: An original version of Schumann's art song in original key (E major), sung by a male voice (tenor).

- `Handel_HWV040-1-01_AR-XXXX.wav`, `Handel_HWV040-1-01_AR-XXXX.wav`: Two arrangement versions of Handel's opera aria *Ombra mai fu* for piano and male voice (tenor), recorded twice in different keys to enable studying the influence of transposition.

- `Brahms_Op115-01_OV-XXXX.wav`: An original version of Brahms' clarinet quintet, 1st movement, played by students currently graduating at secondary school (music excellence branch) or studying at a University of Music, successful at national student-level music competitions.

Beyond enriching our version catalog, these recordings enable to study the difference between systematic, mechanical renderings and real, performed versions of the same work by leaving the acoustic conditions (room and microphone setup) unchanged. We release these recordings under Creative Commons Attribution 3.0 Unported license for research purposes and artistic use.

### A.3 ALIGNMENT EXPERIMENT

To better understand how difficult music synchronization is for the heterogeneous versions of RUBATO and to find the best suitable alignment strategy, we conducted an in-depth study. In the following, we provide detailed results from these experiments complementing Section 3.2 in the paper. The primary objective of this study was to compare traditional signal-processing (SP) features with deep-learning (DL) features under heterogeneous conditions. We conducted the evaluation on the set of audio pairings listed in Table 8, which were specifically selected to represent challenging differences such as instrumentation changes, varying recording quality, and stylistic variation. Examples include aligning Mussorgsky's *The Great Gate of Kiev* as piano version with an orchestral version, or Handel's *Ombra mai fu* realized as a voice with string accompaniment versus an arrangement for accordion ensemble.

The alignment pipeline consists of feature extraction, dynamic time warping and measure transfer. For feature extraction, we compared several types of features: SP chroma features with and without onset refinement, DL-based MPE pitch vectors (`MPE72`) and derived chroma, and hybrid variants combining MPE vectors or chroma with onset information. All experiments were evaluated using median error, 90th and 95th percentile, and normalized area under curve (AUC), quantifying both typical alignment performance as well as robustness against outliers. The score–audio experiments show that learned features generally provide better alignment than SP-based features. While median errors are often comparable across features, DL features are more robust in challenging scenarios. In particular, `MPE_Chroma_Onset` features offer the best balance between precision and robustness for synchronizing both `AR-R` and `OV-R` versions (see Tables 9 and 10).

Audio–audio alignment on pairs with recording-quality differences yielded even stronger results than score–audio alignment when using DL-based features, all achieving comparable results. In

Table 9: Comparison of different features for score–audio (Version 1) alignment across all works. All values are given in ms except AUC (percentage).

| Feature | Median ↓ | 90 perc. ↓ | 95 perc. ↓ | AUC ↑ |
|---|---|---|---|---|
| Chroma | 47 | 194 | 280 | 66 |
| Chroma_Onset | 47 | 186 | 297 | 67 |
| MPE72 | 41 | 147 | 242 | 70 |
| MPE72_Onset | 41 | 158 | 241 | 70 |
| MPE_Chroma | 43 | **151** | 241 | 68 |
| MPE_Chroma_Onset | **39** | 157 | **232** | **72** |

Table 10: Comparison of different features for score–audio (Version 2) alignment across all works. All values are given in ms except AUC (percentage).

| Feature | Median ↓ | 90 perc. ↓ | 95 perc. ↓ | AUC ↑ |
|---|---|---|---|---|
| Chroma | 52 | 219 | 607 | 62 |
| Chroma + Onset | 47 | 206 | 398 | 65 |
| MPE72 | 47 | 314 | 632 | 63 |
| MPE72_Onset | 42 | 216 | 390 | 68 |
| MPE_Chroma | 43 | 206 | 384 | 68 |
| MPE_Chroma_Onset | **41** | **205** | **370** | **70** |

Table 11: Comparison of different features for audio–audio alignment separated in two subsets. All values are given in ms except AUC. Left: Lower quality. Right: Different Instrumentation.

| Feature | Median ↓ | 90 perc. ↓ | AUC ↑ | | Feature | Median ↓ | 90 perc. ↓ | AUC ↑ |
|---|---|---|---|---|---|---|---|---|
| Chroma | 39 | 146 | 69 | | Chroma | 84 | 328 | 47 |
| Chroma_Onset | 37 | 141 | 70 | | Chroma_Onset | 96 | 323 | 44 |
| MPE72 | 34 | **116** | **73** | | MPE72 | 96 | 563 | 52 |
| MPE72_Onset | **33** | **116** | **73** | | MPE72_Onset | 77 | 304 | 51 |
| MPE_Chroma | 36 | 121 | 72 | | MPE_Chroma | **64** | 294 | **56** |
| MPE_Chroma_Onset | **33** | 119 | **73** | | MPE_Chroma_Onset | 69 | **259** | 53 |

contrast, alignment of pairs with substantial instrumentation changes remains considerably more difficult (see Table 11) and was outperformed by score–audio alignment.

Overall, the results indicate that audio–audio alignment on more homogeneous material is easier and more reliable than score–audio alignment. Substantial instrumentation differences, however, remain the most problematic scenario, highlighting the need for specialized alignment strategies in such cases. Across all scenarios, MPE_Chroma and MPE_Chroma_Onset features consistently proved to be the most robust choice for alignment in RUBATO.

### A.4 ANNOTATION TRANSFER DETAILS

To align annotations across versions, we used different multi-stage procedures, as they seem to be optimal for each scenario regarding our findings above. I. e., we consider whether the work is an OV version, an AR with the same instrumentation as its reference (AR-R), or an AR with a different instrumentation.

**OV:**
1. Audio–audio alignment with OV-R using MPE_Chroma features, anchored by manually annotated audio start and end times, to obtain measure and beat positions.
2. Score–audio alignment using MPE_Chroma_Onset features, anchored by beats, to transfer local key, structure and note events (including transposition where required).

**AR (same instrumentation as AR-R):**
1. Audio–audio alignment with AR-R using MPE_Chroma features, anchored by manually annotated audio start and end times, to obtain measure positions.
2. Audio–audio alignment with OV-R using MPE_Chroma features, anchored by measures, to obtain beat annotations.
3. Score–audio alignment using MPE_Chroma_Onset features, anchored by beats, to transfer local key, structure and note events (including transposition where required).

Table 12: Overview of the folder structure of RUBATO

| Folder Name | Content Description |
|---|---|
| - 01_RawData | |
| \|- score_image | PNG image exports from the symbolic score |
| \|- score_midi | MIDI exports from the symbolic score |
| \|- score_musescore | Symbolic Score in MuseScore format |
| \|- score_pdf | Symbolic Score in PDF format |
| \|- video_30fps | Video recordings of the piano performance corresponding to the MIDI file of the score |
| \|- wav_22050_mono | Audio files resampled |
| - 02_Annotations | |
| \|- ann_audio_beat | Beat annotations given in physical time and frames |
| \|- ann_audio_localkey | Local key annotations given in physical time |
| \|- ann_audio_measure | Measure annotations given in physical time and frames |
| \|- ann_audio_noteEvents | Note Events with start and end given in physical time, velocity and instrument |
| \|- ann_audio_startEnd | Start and end times of tracks |
| \|- ann_audio_structure | Structure annotations given in musical time |
| \|- ann_score_localkey | Local key annotations given in musical time |
| \|- ann_score_structure | Structure annotations given in musical time |
| - 03_ExtraMaterial | |
| \|- scripts | Python scripts for preprocessing and alignment |
| \|- warpingPaths | Warping paths aligning each audio track to its corresponding symbolic score |

**AR (different instrumentation than AR-R):**

1. Score–audio alignment using MPE_Chroma_Onset, anchored by manually annotated audio start and end times, to transfer all annotations (including transposition where required).

## A.5 USING RUBATO

### A.5.1 DATASET STRUCTURE

Following the practice described by Weiß et al. (2021), we organize the data in RUBATO into a systematic folder structure. Table 12 provides a detailed overview of all folders and subfolders. Within each folders, we name all score-related files as ComposerID_WorkID.ext and all audio-related files as ComposerID_WorkID_VersionType-VersionID.ext with .ext denoting the respective file extension. We store all annotations in a standardized tabular format (.csv), listing measure labels alongside time information.

### A.5.2 PROPOSED TRAINING–VALIDATION–TEST SPLIT

As described in Section 4, we propose to use the entire RUBATO dataset as an unseen benchmark dataset for testing models trained on considerably larger dataset. Nevertheless, it may be interesting to exploit the particular nature of RUBATO's data also for training, fine-tuning, or regularizing (using the cross-version consistencies) DL models, or for conducting domain adaptation to this data.

As discussed in Weiß & Peeters (2022), however, there are caveats when splitting dataset to still obtain results that are representative for real-world performance. Structured multi-version datasets, where each work is present in the same set of versions, allow for splitting along two axes, the work and the version axis. Enforcing models' generalization across both axes leads to the so-called "neither split," where neither the work nor the version characteristics of each test track have been seen by the model before. This is not possible for RUBATO, as (exept for SR and SY versions), we have a unique set of versions for each work. Consequently, each work split is also a neither split when taking care of the SR and SY versions. Since music analysis and transcription models are known to overfit to specific works, we always propose to always use such a work split

Considering, in addition, to balance instrumentation, mode (major/minor), tempo, and style as much as possible, we therefore propose the following standard best-practice splitting strategy for RUBATO:

- **Test set**: All versions except the synthesized SY-EWSO versions of the following works: Mozart_KV618, Schumann_Op039-05, Mussorgsky_Pict-10, Brahms_Op115-01.
- **Validation set**: All versions except all SR and SY versions of the following works: Beethoven_Op047-01, Handel_HWV056-2-44.

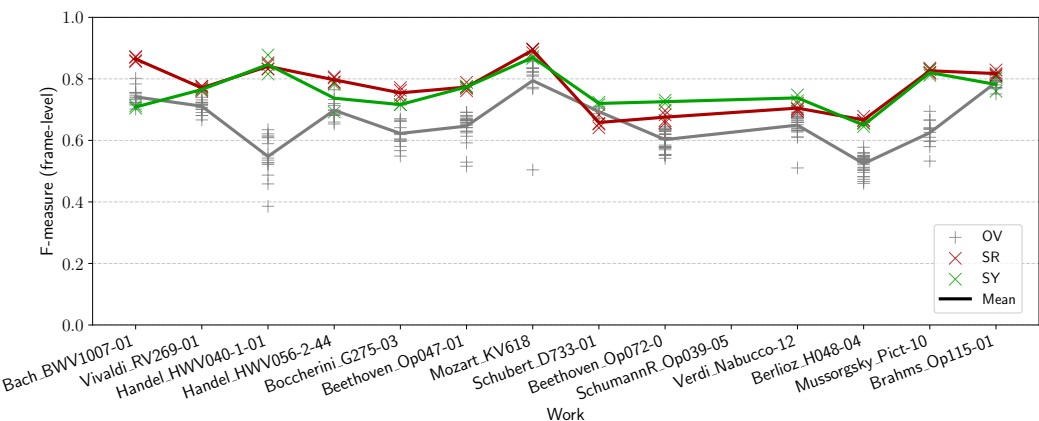

Figure 7: Frame-level AMT results per work for `ResNet`.

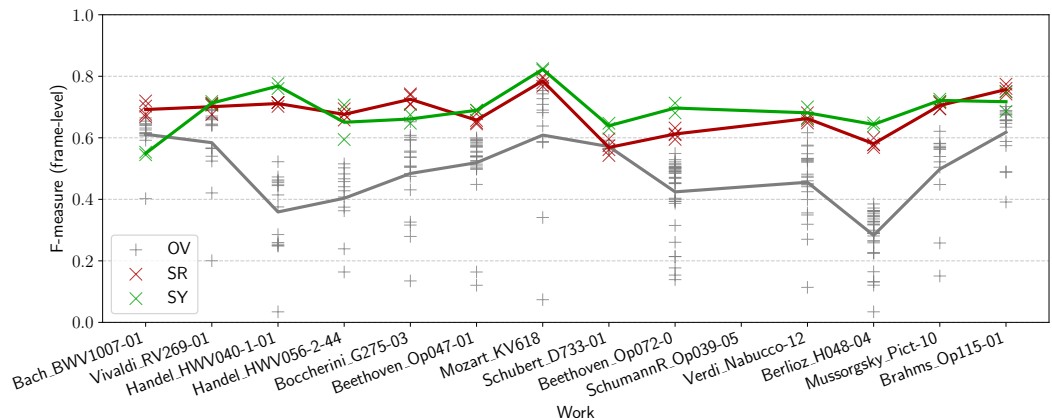

Figure 8: Frame-level AMT results per work for `BP-NOTE`.

- **Training set**: All versions except all `SR` and the synthetic `SY-EWSO` version of the remaining six works.

Beyond the considerations above, this guarantees that the test set comprises unseen composers only. Moreover, all systematic recordings taken in the studio are used for testing only, and the test set allows to calculate all pairings for the cross-version consistency as used in the unseen benchmark (Section 4). Synthesized recordings are split between training and test set according to the different sample libraries used.

## B   RUBATO BENCHMARK, FURTHER RESULTS

To complement the detailed per-work results for the AMT benchmark (Section 4.2), we present further results for the frame-wise MPE task for selected other models:

- `ResNet` (Figure 7),

- `BP-NOTE` (Figure 8),

- `ReconVAT` (Figure 9), and

- `YMT3+` (Figure 10).

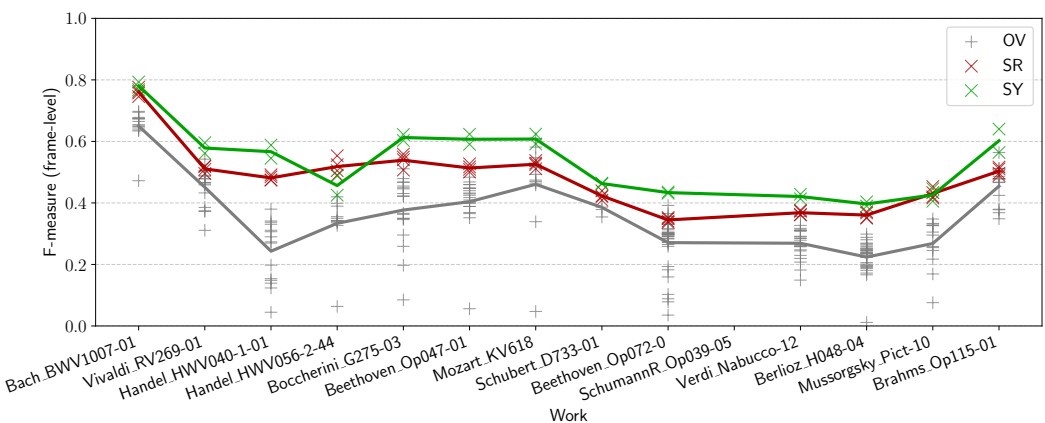

Figure 9: Frame-level AMT results per work for `ReconVAT`.

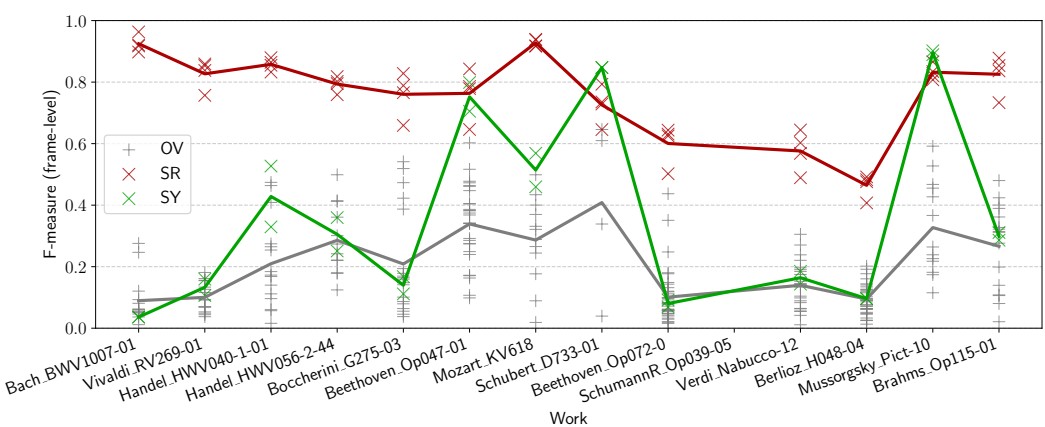

Figure 10: Frame-level AMT results per work for `YMT3+`.

