# OpenReview forum: "RUBATO: A Multi-Version Benchmark for Robust Music  Analysis and Transcription"
_ICLR.cc/2026/Conference — ICLR 2026 Conference Withdrawn Submission_

### Official Review · Reviewer_27TR · 2025-10-31

**Soundness:** 3
**Presentation:** 4
**Contribution:** 2
**Rating:** 4
**Confidence:** 3

**Summary:**

The paper presents a benchmark dataset, RUBATO, that is specialized for multi-version related tasks on classical music. The authors have put significant effort creating this dataset including recording new versions. The data consists of carefully chosen works and versions, which are aligned semi-automatically. The datasets also includes expert labels of the structure, pitch, instruments, key, beats, and chords.

**Strengths:**

The paper and the proposed dataset definitely seem to be of high quality. As far as I can tell, many musicological aspects are considered in the design of the dataset. Multi-version is a unique, interesting aspect of classical music and the proposed dataset would significantly help evaluating the aspect.

**Weaknesses:**

* Title: "Robust" is a much more general term than how it is used in this paper. I do think the proposed dataset provides a way to test robustness of models, but "Robust Music Analysis" sounds too general when what it actually means is a cross-version consistency.
* Section 4: RUBATO Benchmark: These experiments are interesting, but I disagree with some of the interpretations. I find the conclusion of "This suggests that medium-sized, fully convolutional models obtain slightly lower efficacies but are more robust and, therefore, universally applicable than large Transformers." (on L 426-427) a bit premature. The models are different from various aspects and we can't simply attribute the different behaviors to the difference of model architectures.
* Impact: The proposed dataset is unfortunately limited to 1) evaluation of 2) classical music, and especially on 3) the consistency between versions. I firmly believe it took a significant amount of effort and consideration to build this dataset, and for the purpose it would be a great dataset. But that seems really narrow, otherwise I would expect the consistency related discussion in 4.2 Result - the last 5 lines - would have deeper discussion. However, that discussion is rather simple and a bit questionable. Similarly, I am not convinced about the discussion in Section 5 (more details on Questions.)

**Questions:**

In Section 5 - Results:
- On "Figure 4 shows the key recall rate for Octave-LSTM. Efficacy is generally high as compared to a typical inter-annotator agreement.": I find it a bit confusing that the authors discuss the performance of the model by comparing it with inter-annotator agreement. To be strict, the efficacy being high or low has nothing to do with inter-annotator agreement. Wouldn't it make more sense to compare GEC or other CVC measures with a typical inter-annotator agreement?
- The key detection and transcription would have very different random prediction baselines, and they're essentially very different tasks. This would make all the CSC not comparable across tasks.
- On "This suggests that for LKE, the challenge is less the variety of versions but to learn the musical notion of local key": I kind of get it, but not fully. It is as if implying there are only two aspects of solving a task: inter-version variety or the core task itself? And can we actually separate them? Isn't "to learn the musical notion of local key" == "be able to do it with different styles (that are present in different versions)" -- i.e., isn't the latter include the former?
  - Would it be the case that, as the model prediction score goes higher, in reality LEC and LPC would go higher as well? I'm asking this question as I wonder if we could simply conclude "LKE is a relatively easy task and both of the models perform quite consistently well." In other words, the cross-version difference might be still challenging as much as the problem is, and we can't really argue if the challenge is more or less the variety of versions vs. learn the musical notion of local key."

---

> ### Author Response · Authors · 2025-11-27
>
> First of all, thank you for your valuable feedback!
>
> Regarding the term robustness: We propose CVC as a set of diagnostic research tools that, by design, measure sensitivity to timbral and acoustical variations. We strongly believe that this sensitivity captures a substantial part of what robustness in music audio analysis means. While “robustness” is indeed a broad term, CVC is one concrete and important facet of it-Analogous to the many facets of robustness that have been studied extensively in the image domain.
>
> Regarding results in Section 4.2: We agree that, with the currently small number of evaluated pretrained models (all trained on different datasets), our results do not fully support the conclusion regarding robustness differences between Transformers and convolutional models. Our intention was to present preliminary findings and instructive examples of how the benchmark can be used rather than definitive claims. In the revised version, we will clarify this point. We are currently working on more experiments to better understand the effects of architectures and training frameworks on model robustness.
>
> To clarify our discussion on inter-annotator agreement (Section 4.3): Since the key recall rate for Octave-LSTM has surpassed the thresholds of typical inter-annotator agreement, we cannot tell whether any improvement on standard metrics is valid or merely reflects overfitting to a single annotator’s preferences. Therefore, using CVC as an additional evaluation metric is highly valuable, complementing metrics such as key recall rate.
>
> Regarding comparability between tasks: It is correct (and particularly true for music analysis tasks) that evaluation is not easily comparable across tasks. However, this issue applies to all other evaluation measures as well.
> Regarding the two aspects of the task: Cross-version variety and the core task itself can be correlated but can also be separated. Our purpose in using CVC is to isolate the challenge of cross-version variety as a measure of a model’s robustness. From a more general machine learning perspective, robustness can be included in measuring how "good" a model is, but it can also be separated so that we are able to see how well the model learns the core task and how stable it is under other variances of the data. We do not say that both models perform consistently well, because it is clear that Octave-LSTM is both better in efficacy and higher in consistency. Our argument for a lower cross-version challenge in LKE is based on comparison with AMT, where we observe clear challenges when generalizing from piano to original versions. We will make this comparison more explicit in a future version of our manuscript. For LKE itself, we currently have only two models, one of which performs well. Consequently, we are not able to determine whether the main challenge lies in the variety of versions or in learning the musical notion of the task (though our intuition points to the latter).

---

### Official Review · Reviewer_UPwR · 2025-11-01

**Soundness:** 2
**Presentation:** 3
**Contribution:** 2
**Rating:** 4
**Confidence:** 4

**Summary:**

RUBATO is a multi-version dataset and benchmark for evaluating the robustness of music analysis and transcription models. The dataset comprises 14 canonical works with up to 54 versions each (560 tracks, 42 hours), including original recordings, arrangements, controlled piano renderings, and synthesized audio. A key contribution is the provision of high-quality, structurally coherent annotations, transferred across versions using a rigorously evaluated alignment strategy. Beyond standard evaluation, the authors propose Cross-Version Consistency (CVC) metrics (GEC, LEC, LPC) to quantify model robustness. It demonstrated significant robustness gaps in current state-of-the-art models through experiments on Automatic Music Transcription (AMT) and Local Key Estimation (LKE).

**Strengths:**

The dataset is  constructed. The systematic inclusion of different version types (original, arranged, controlled) enables targeted studies on specific domain shifts. The effort to ensure structural coherence and the rigorous alignment process are commendable. It is an open resource dataset with clear structure and the formalization of CVC provides a valuable, additional lens for evaluation beyond standard accuracy, better reflecting the requirements for real-world, musicologically-grounded applications.

**Weaknesses:**

A significant omission is the evaluation of large, generative, or foundation models in music. The benchmark is limited to established, discriminative models for AMT and LKE (e.g., MT3, Basic Pitch). The field is rapidly moving towards large-scale models and instruction-tuned models capable of multiple tasks (e.g., Qwen2.5 omni, Gemini). Perhaps ABC notation can also be provided for this. The benchmark's relevance is diminished by not including such models. For instance, it is unclear how these models would perform on tasks like version retrieval, structure analysis, or zero-shot transcription across RUBATO's heterogeneous versions. This limits the paper's impact on the current research frontier.

The exclusive focus on 18th- and 19th-century Western classical music, while practical for copyright reasons, constrains the generalizability of the findings. Robustness challenges in pop, jazz, or non-Western music may differ significantly.

**Questions:**

The benchmark currently evaluates single-task, discriminative models (e.g., MT3). However, the field is rapidly advancing towards large, multi-task audio foundation models capable of both transcription and analysis, often in a generative or instruction-following setting (e.g., "transcribe the violin part," "describe the musical structure"). In this new context, how do the authors envision the RUBATO benchmark evolving? Specifically: 1) How can the CVC metrics be adapted to evaluate the consistency of a single, multi-task model's outputs across versions, as opposed to comparing dedicated, single-task models? 2) Does the concept of a fixed, frame-level output representation (e.g., a 72-bin pianoroll) become a limitation when evaluating a model that might natively output a symbolic score, a text description, or a combination thereof?

The paper's tasks and annotations are firmly grounded in the paradigm of Western Common Practice notation, where discrete "notes" and a fixed set of "keys" are fundamental concepts. So how do the authors view the generalizability of their proposed evaluation metrics (e.g., F-measure on note events, MIREX score for key) to musical cultures where the very concept of a "note" is different (e.g., Raga music, genres with continuous pitch slides) or where the perceptual foundation of harmony does not align with the Western key system?

Also, given that robustness is a universal goal, what steps could be taken in the future to extend the RUBATO framework—or to inspire a similar benchmark—that is agnostic to a specific notational culture, perhaps relying more on perceptual similarity or culture-specific ground truths?

**Details Of Ethics Concerns:**

No.

---

> ### Author Response · Authors · 2025-11-27
>
> First of all, thank you for your valuable feedback!
>
> Regarding the use of foundation models: We agree that this would be an interesting addition. There are music-understanding benchmarks specifically for multimodal models (e.g., MuChoMusic). To the best of our knowledge, the musical understanding of multimodal foundation models is (yet) not comparable to models specialized for these tasks. For this reason, we decided to exclude them from the benchmark. However, we expect that this field is moving forward quickly and, therefore, plan to evaluate such foundation models on the RUBATO benchmark in future work.
>
> Regarding the evaluation of multi-task models: We do not see a problem in using CVC for multiple tasks for a single multi-task model. Using your example, we can compute CVC for the violin transcription and compute CVC for the structure annotations individually. However, as you correctly point out, in its current design, LPC and LEC depend on a representation that can be mapped to time frames. The 72 bins are a choice for evaluating this specific task—the RUBATO dataset as well as the CVC framework are not limited to this decision but allow for comparing arbitrary vectors. Yet, adapting the framework for textual outputs without clear timestamps is rather challenging. Even though it is an intriguing idea, there are currently no plans to move in that direction.
>
> Regarding the evaluation of discrete notes and keys, we want to emphasize that the general idea is transferable as well, yet of course not with the Western classical music data provided here. Which characteristics are belonging to the “work” and which ones to the “version” (performance) depends on the musical genre. Consider performances of jazz standards, for example, where the rough harmonic progression (up to re-harmonizations) and the main melodic theme are the consistent “work” characteristics, while the remaining characteristics (improvisations over this general structure) are unique to each “version”. The great opportunity of Western classical music is that many characteristics (pitch, onset, harmony, rhythm, structure, …) are belonging to the “work”, and that these characteristics are explicitly written down in the score, which lets us derive high-quality annotations.
>
> Regarding the limitation to Western classical music: For this dataset, we exploit a feature that is unique to Western classical music: the abundance of multiple versions with the exact same musical content and the existence of scores that musicians closely follow. Therefore, CVC cannot be extended to other genres such as pop, jazz, or non-Western music without limiting the analysis dimensions to global tasks (e.g., work identification or global key estimation) or ill-defined problems (e.g., genre recognition). Generalizability of the findings beyond Western classical music is still an open question. However, we believe that sensitivity to timbral diversity between versions is a concept that exists in all genres, and that a model that is robust to such variations on RUBATO will likely be robust in other genres as well. In this way, the methodological insights are highly likely to apply to other music scenarios—Western classical music simply provides a unique opportunity for systematic studies.

---

### Official Review · Reviewer_1f3Z · 2025-11-01

**Soundness:** 2
**Presentation:** 3
**Contribution:** 2
**Rating:** 4
**Confidence:** 4

**Summary:**

This paper introduces RUBATO, a new multi-version benchmark designed to evaluate robustness in music audio analysis and transcription.
It leverages Western classical works that share the same musical score but vary across performances, instrumentations, and recording conditions, providing a systematic basis to test model generalization.

RUBATO contains 560 recordings (42 hours) of 14 canonical works by 12 composers, covering original performances, arrangements, controlled piano renderings, synthesized versions, and adaptations—all aligned to a shared musical time axis through a validated hybrid alignment pipeline. The dataset ensures structural coherence across most versions and includes annotations sufficient for evaluating diverse music analysis tasks.

Beyond the dataset, the paper proposes the RUBATO Benchmark, introducing Cross-Version Consistency (CVC) metrics that quantify how stable a model’s predictions remain across versions of the same work. Using these metrics, the authors evaluate state-of-the-art systems for Automatic Music Transcription and Local Key Estimation, showing that while these models perform well on individual recordings, they often fail to generalize consistently across different versions—revealing a clear robustness gap in current music models.

**Strengths:**

S1:Well-engineered and reproducible data pipeline.The dataset construction is technically solid and clearly described, including systematic collection, controlled re-recordings, MIDI-based renderings, unified formatting, and a multi-stage alignment process (manual anchors → audio–audio → score–audio). Quantitative alignment error analysis adds credibility and reproducibility.

S2:Clear formalization of cross-version evaluation.The paper explicitly defines robustness as Cross-Version Consistency (CVC) and decomposes it into GEC, LEC, and LPC. This makes robustness measurable and avoids conflating accuracy with stability.

S3:Alignment quality is quantitatively validated.Synchronization accuracy is reported with concrete AUC values and human-level tolerance references, demonstrating methodological rigor and providing realistic expectations of annotation noise.

**Weaknesses:**

W1:

The overall idea—evaluating robustness through cross-version consistency—is conceptually reasonable but not new. Similar robustness and consistency frameworks have been widely explored in other domains since around 2019–2022 (e.g.,*ImageNet-C*,*WILDS*,*BIG-bench*).What this paper mainly contributes is an adaptation of those ideas to the music domain, without introducing genuinely new methods or theoretical insights.

W2: While the three proposed measures (GEC, LEC, LPC) are well defined, the paper never demonstrates that they truly capture robustness. There is no evidence that these metrics correlate with human perception, alignment reliability, or downstream generalization. The authors even acknowledge that a model could score high on CVC while being consistently wrong, which raises serious doubts about the interpretability and usefulness of the metric itself.

W3: Only two tasks—Automatic Music Transcription and Local Key Estimation—are evaluated, although the dataset contains annotations for several others (e.g., beat tracking, structure analysis). This makes it difficult to justify RUBATO as a general benchmark for “robust music analysis.”

W4: The dataset is limited to Western classical music, which simplifies alignment and annotation but prevents generalization to popular, modern, or non-Western styles where variability is higher. The scope is narrow and misses the opportunity to test robustness in more diverse musical conditions.

W5: The observation that Transformer-based models are less consistent than convolutional ones is interesting but remains purely descriptive. The paper does not explore potential causes such as timbre sensitivity, training-domain bias, or decoding instability.

W6:  All experiments are run on small, task-specific architectures. It remains unclear whether similar robustness issues would appear in large pretrained or foundation models, which are increasingly standard in audio and MIR research.

W7:  Cross-version consistency analysis is unlikely to be used in real-world music applications. At present, it serves mainly as a diagnostic research tool rather than an applied benchmark. The paper doesn’t convincingly explain how the proposed metrics or dataset would translate to practical or industrial benefit.

W8：From a methodological and temporal perspective, the contribution feels incremental and somewhat out of step with where robustness research has moved. The work will likely be useful for the MIR community as a reproducible resource, but the contribution remains incremental rather than forward-looking.

**Questions:**

Q1: The dataset includes annotations for other MIR tasks such as beat tracking and structure analysis, yet the experiments focus only on AMT and LKE.
Could you clarify whether the proposed CVC framework can be meaningfully extended to these additional tasks, and if so, what limitations or adjustments would be required?
Q2: The finding that Transformer-based models show lower cross-version consistency than convolutional ones is intriguing, but the discussion remains quite descriptive.
Could you provide further quantitative or qualitative analysis to explain the source of this instability—such as the influence of timbre sensitivity, domain bias, or decoding thresholds?
Q3: Have you considered whether recent music-pretrained representation models—such as MERT—exhibit similar cross-version consistency patterns at the representation level?
Even if these models are not designed for low-level transcription, testing them could help determine whether the robustness patterns you report also appear in higher-level semantic or structural representations.

---

> ### Author Response · Authors · 2025-11-27
>
> First of all, thank you for your valuable feedback!
>
> Response Q1: Regarding the global measure GEC, there is no limitation, as it can be used for all tasks (as long as a global evaluation measure exists). The way the local measures (LEC, LPC) are currently formulated, they work for all tasks that can be viewed as framewise. This includes beat tracking (possibly accounting for small alignment inaccuracies by applying suitable tolerance thresholds).
>
> Response Q2: These are still open questions as our intention was to present preliminary findings and instructive examples how the benchmark can be used. For now, we just observe that transformer-based models seem to struggle more on unseen datasets (also noted in [1], Table 5). From our experience, instrument-dependent thresholds do seem to play a role. We will explore this in more depth in future work.
>
> Response Q3: We have not conducted any experiments with foundation models. A limitation here is that these representations (such as MERTs) are not version-invariant, meaning they capture by design a mixture of version-specific (timbre, sound, instruments, performance style) and version-independent (pitch, rhythm, structure) characteristics. However, probing them with supervised classification heads may enable studying their version-independent characteristics. We are thankful for the comment and plan to explore this in future work.
>
> Regarding W2: We propose CVC (GEC, LEC, LPC) as a set of diagnostic research tools that, by design, measure sensitivity to timbral changes and treat this as one definition of robustness. Furthermore, previous studies have shown a correlation between LPC and generalization capabilities (i.e., performance on out-of-domain data) [2]. In these experiments, we also found that the trivial case of models being consistently wrong does not play a role in practice, as we view CVC as an additional figure of merit that supplements common evaluation.
>
> Regarding W4: For this dataset, we exploit a feature unique to Western classical music: the abundance of multiple versions with the exact same musical content and the existence of scores that musicians closely follow. Therefore, CVC cannot be extended to other genres such as pop, jazz, or non-Western music without limiting the analysis dimensions to global tasks (e.g. work identification or global key estimation) or ill-defined problems (e.g. genre recognition). Generalizability of the findings beyond Western classical music remains an open question. However, we do believe that sensitivity to timbral diversity between versions is a concept that exists in all genres, and that a model robust to such variations on RUBATO is likely to be robust in other genres as well. In this way, the methodological insights are highly likely to apply to other music scenarios – Western classical music simply provides a unique opportunity for systematic studies.
>
> [1] Josh Gardner, Ian Simon, Ethan Manilow, Curtis Hawthorne, and Jesse H. Engel. MT3: multi-task multitrack music transcription. In Proc. 10th International Conference on Learning Representations (ICLR), 2022. In Proc. 10th International Conference on Learning Representations (ICLR), 2022.
>
> [2] Yannik Venohr, Yiwei Ding, and Christof Weiß. Towards robust music transcription by measuring cross-version consistency in Western classical music. In Proc. International Society for Music Information Retrieval Conference (ISMIR), 2025.

---

### Note · Authors · 2026-01-12

I have read and agree with the venue's withdrawal policy on behalf of myself and my co-authors.